# Importance of Asprosin for Changes of M. Rectus Femoris Area during the Acute Phase of Medical Critical Illness: A Prospective Observational Study

**DOI:** 10.3390/healthcare11050732

**Published:** 2023-03-02

**Authors:** Hilal Sipahioglu, Hatice Kubra Zenger Ilik, Nurhayat Tugra Ozer, Sevda Onuk, Sumeyra Koyuncu, Sibel Kuzuguden, Gulseren Elay

**Affiliations:** 1Department of Intensive Care Unit, Kayseri City Training and Research Hospital, Kayseri 38080, Turkey; 2Department of Internal Medicine, Kayseri Training and Research Hospital, Kayseri 38080, Turkey; 3Department of Clinical Nutrition, Erciyes School of Medicine, Erciyes University, Kayseri 38080, Turkey; 4Department of Nephrology, Kayseri City Training and Research Hospital, Kayseri 38080, Turkey; 5Department of Biochemistry, Kayseri City Training and Research Hospital, Kayseri 38080, Turkey; 6Department of Intensive Care Unit, Gaziantep University, Gaziantep 27470, Turkey

**Keywords:** asprosin, adipokine, older adult patients, muscle mass, ICU

## Abstract

Asprosin, a new adipokine, is secreted by subcutaneous white adipose tissue and causes rapid glucose release. The skeletal muscle mass gradually diminishes with aging. The combination of decreased skeletal muscle mass and critical illness may cause poor clinical outcomes in critically ill older adults. To determine the relationship between the serum asprosin level, fat-free mass, and nutritional status of critically ill older adult patients, critically ill patients over the age of 65 receiving enteral nutrition via feeding tube were included in the study. The patients’ cross-sectional area of the rectus femoris (RF) of the lower extremity quadriceps muscle was evaluated by serial measurements. The mean age of the patients was 72 ± 6 years. The median (IQR) serum asprosin level was 31.8 (27.4–38.1) ng/mL on the first study day and 26.1 (23.4–32.3) ng/mL on the fourth study day. Serum asprosin level was high in 96% of the patients on the first day, and it was high in 74% on the fourth day after initiation of enteral feeding. The patients achieved 65.9 ± 34.1% of the daily energy requirement for four study days. A significant moderate correlation between delta serum asprosin level and delta RF was found (Rho = −0.369, *p* = 0.013). In critically ill older adult patients, a significant negative correlation was determined between serum asprosin level with energy adequacy and lean muscle mass.

## 1. Introduction

Per the results of the world population estimates revised in 2017, Europe faces exceptional demographic changes. Accordingly, individuals aged 60 and above already constitute 25% of the population, and by 2050, this ratio is expected to reach 35%. The number of those with an age of 80 and above will be tripled by 2050 [1]. In developed countries, the increased average life expectancy has also resulted in increased demand for hospitalization of the elderly population in hospitals and intensive care units (ICU). It has been determined that approximately more than 50% of the patients hospitalized in the intensive care unit are above 65 years old [2,3,4]. A systematic review demonstrated a significantly high malnutrition prevalence (38–78%) in ICU patients. This situation is correlated to an increase in morbidity, mortality, and hospital-related costs for patients [5].

The dependency on mechanical ventilation is correlated with malnutrition, length of hospital stay, ICU readmission, infection rates, and risk of hospital death, making this a critical dilemma in ICU patients’ care. There are significant challenges in accurately estimating energy requirements and hence the optimal dosing of nutrition. Critical illness results in hypermetabolism and hypercatabolism, putting patients in the ICU at high risk of malnutrition. The metabolic and hormonal changes in critical illness result in muscle wasting and associated ICU-acquired weakness, which can persist for years [6].

Muscle atrophy can occur relatively early in critically ill patients in intensive care units. Muscle atrophy occurs with increased destruction and decreased muscle protein synthesis [7,8]. Inflammation, immobilization, endocrine stress responses, rapidly developing nutritional deficit, impaired microcirculation, and denervation are conditions that accelerate muscle atrophy [9,10]. Additionally, muscle loss is common in humans due to aging. Accordingly, muscle loss caused by aging may deepen in the presence of critical illness. Reversing skeletal muscle catabolism can prevent muscle atrophy during critical illness and improve functional outcomes [11,12,13]. Proinflammatory mediators are used as an indicator of muscle atrophy during critical illness [14]. Ultrasound is widely used in clinical practice, greatly contributing to diagnosis and management of many conditions. While systematic ultrasound examinations have been conducted mainly by sonographers in an examination room, there is now considerable interest in having physicians perform ultrasounds at the bedside, as part of regular medical examinations. Studies using portable ultrasounds have been spreading not only in the emergency room and intensive care unit (ICU) settings, but also in out-of-hospital situations in, for example, primary care and long-term care facilities (e.g., nursing homes). Additionally, muscle ultrasound is a suitable method for evaluating patients with muscle atrophy. The ultrasonographic evaluation of quadriceps’ muscle thickness effectively determines the effect of nutritional interventions on muscle loss in critically ill patients [7,8].

Adipose tissue functions as an endocrine organ with central energy storage that creates a diversity of bioactive mediators and adipokines (adipose-derived secreted factors), possessing proinflammatory or anti-inflammatory impacts. Adipokines may easily move into the systemic circulation and perform their effects through an inter-cell communication network (autocrine, paracrine, endocrine). Furthermore, they preserve regulating several aspects of the normal metabolic processes in the human body, such as glucose and lipid homeostasis, insulin sensitivity, and inflammatory response [15]. Asprosin is a novel glucogenic adipokine discovered in 2016, mainly secreted from white adipose tissue, and has a critical role in the regulation of hepatic glucose release, insulin secretion, appetite, and inflammatory response [16]. Moreover, it activates the PKCδ/SERCA2-mediated endoplasmic reticulum stress/inflammation pathways in skeletal muscle and promotes insulin resistance [17]. Insulin resistance has been revealed to be relatively higher in critically ill patients compared to healthy patients [10]. Evidence suggests an association between asprosin secreted levels and weight loss extent as a result of bariatric surgery, including sleeve gastrectomy or cholecystectomy. Two studies indicated a significant decrease in serum asprosin levels after six months of weight loss surgical intervention [18,19]. During fasting, the circulating serum asprosin level rises according to the glucose requirement and decreases with the start of feeding. Providing adequate nutritional support to critically ill patients has a critical role in the clinical prognosis of the patient [20,21]. Considering the above-mentioned information, the relationship between asprosin, muscle mass, and nutritional support in critically ill elderly patients is unclear.

This study aimed to reveal the relationship between the serum level of asprosin, a new adipokine, the change in lean muscle mass in critically ill elderly patients, and the nutritional support given to the patients.

## 2. Material and Methods

### 2.1. Study Design and Participants

This study presents a prospective observational design developed in a tertiary care hospital’s clinical-internal intensive care unit from March to September 2022. All patients over the age of 65 who were expected to stay in the intensive care unit for at least 4 days and were administered enteral nutrition support within the first 48 h after their admission to the ICU were included. Patients who could be fed orally, who had previously been treated with parenteral therapy, and who had contraindications for enteral nutrition were excluded from the study.

The study was approved by the local ethics committee (No: 586, date: 24 February 2022) and was conducted according to the Helsinki Declaration guidelines. Free and informed consent was obtained from the legal guardians of the study participants.

### 2.2. Data Collection

Demographic data, ICU admission diagnostics, comorbidities, APACHE II (Acute Physiology and Chronic Health Evaluation) scores, SOFA (Sequential Organ Failure Assessment) scores, and the Charlson comorbidity index were recorded at admission.

During the follow-up, the need for a mechanical ventilator, renal replacement requirement, and the number of days spent in the intensive care unit and hospital were recorded.

The energy target of the patients was calculated as 25–30 kcal/kg/day according to ESPEN Recommendations [20]. The daily energy intake of the patients by tube who actually received enteral nutrition for four days from the start of enteral nutritional support was recorded. The percentage of patients reaching the target energy was calculated. No adjustments were made for age or BMI when calculating energy targets. The risk of malnutrition in patients was determined by the NRS-2002 score. The NRS-2002 form was filled in by the nurses on the day that the patients were admitted to the intensive care unit, taking information from the patients and their relatives, and recorded in the patient file. The nutritional risk of the patients was determined, and a nutrition plan was made. Patients with NRS-2002 ≥ 3 were considered to be at risk for malnutrition.

### 2.3. Serum Asprosin Measurement

Here, 3 mL blood samples were collected in tubes, and the samples were centrifuged at 3000× *g* for 10 min at the 24th hour (first day) and fourth day after the start of enteral nutrition support. In our intensive care unit, feeding is interrupted at 11 am for all patients receiving enteral nutrition. Blood samples were drawn in the morning fasting before feeding was re-initiated. Then, 1 mL of serum supernatant was removed and collected in an Eppendorf tube. Serum samples were kept frozen at −80 °C. Serum asprosin protein concentrations were analyzed using the ELISA method (Cat. No. E4095HU). Delta (Δ) asprosin was calculated as the difference between the first and fourth day asprosin levels of the patients. The normal asprosin level was considered as <23.6 ng/mL (according to the reference range (kit used) determined by the Bioassay Technology Laboratory).

### 2.4. Ultrasonographic Assessment

Ultrasound measurements were performed at 24 h (day 1) and on day 4 after the start of enteral nutritional support. Philips ClearVue 550 system with a linear ultrasound probe (4–12 Mhz) was used for calculation while these measurements were in the supine position on the surface in B mode. The area of the rectus femoris (RF) muscle of the lower extremity quadriceps muscle was measured. The sensor was perpendicular to the thigh axis, and the point is located at 2/3 of the distance from the anterior superior iliac spine to the upper border of the patella.

All ultrasonography (USG) measurements were performed by an intensive care specialist with five years of USG experience. Delta (Δ) RF was calculated as the difference between the RF area of the patients on the first and fourth days.

### 2.5. Statistics

Statistical analysis was performed using the IBM SPSS statistics program version 22 (IBM, New York, NY, USA). The normality distributions of continuous variables were examined using the Shapiro–Wilk test. According to the normal distribution, continuous variables were presented as mean ± SD or median (interquartile range). Categorical variables were shown as numbers (%, percentage). The correlation between the data was investigated using Spearman’s correlation test. The correlation coefficient was accepted as 0–0.29 (weak), 0.30–0.69 (moderate), and 0.70–1.0 (strong). A value of *p* < 0.05 was considered statistically significant for all analyses.

The study sample size was calculated as 42 patients, with a medium effect size according to the baseline asprosin level (d = 0.5), 80% strength, and 5% error probability using G-Power 3.1 software.

## 3. Results

Two hundred and fifty-one patients hospitalized in the intensive care unit were evaluated. Of these patients hospitalized in the intensive care unit, 94 were under the age of 65, and 94 did not receive enteral nutrition (26 who received oral nutrition, 68 who received parenteral nutrition) were excluded. First, 67 patients were included in the study. However, 12 patients died during the study period, and 7 patients were excluded since their serum blood was hemolyzed. Two patients were excluded from the study because they switched from enteral to oral feeding. As a result, a total of 46 patients were included in the study (Figure 1).

The mean age of the patients was 72 ± 6 years, and the median value of males (IQR) was 25 (54.3). The median (IQR) BMI of the patients was 22.0 (20.9–29.0), the mean APACHE II was 19.8 ± 6.98, and the median (IQR) Charlson comorbidity index was 6 (4–8). Metabolic disorders (*n*: 17, 37.0%) and sepsis/septic shock (26.1%) were the most common reasons for hospitalization in the intensive care unit. Diabetes mellitus was present in 16 (34.8%) of our patients, and at the same time, all the patients had received insulin therapy. The most common comorbidity in our patients was hypertension, in 29 patients (63.0%). The malnutrition risk was found in 32 patients (70.0%). The median percentage of reaching the daily energy requirement was 65.9 ± 34.1 during the 4-day follow-up. The daily energy requirements and the amount they can actually take are presented in Table 1. The mean daily protein intake was 0.4 ± 0.27 g/kg/day on the first day and 0.8 ± 0.47 g/kg/day on the fourth day.

Mechanical ventilation (21 (45.7%)) and renal replacement requirement (21 (45.7%)) of the patients were quite high. Demographic data and clinical characteristics of the patients are presented in Table 1.

Median (IQR) asprosin levels were 31.8 (27.4–38.1) ng/mL on the first day and 26.1 (23.4–32.3) ng/mL on the fourth day. The serum asprosin concentration of study participants significantly decreased (*p* < 0.001), and the delta asprosin value was −5.77 (−9.22 to 0.28) (Table 2).

The asprosin level was high in 95.7% of the patients on the first day. This rate decreased on the fourth day of the study, and 73.9% of patients had high asprosin levels (Figure 2).

Median RF was 1.68 (1.35–2.07) cm^2^ on the first day and 1.82 (1.38–2.01) cm^2^ on the fourth day (*p* = 0.196). The median delta RF was 0.15 (−0.43 to 0.46). The laboratory values of the patients on the first and fourth days are presented in detail in Table 3. The glucose level of the patients was 143 (110–194) mg/dL on the first day, 125 (103–170) mg/dL higher than on the fourth day (*p* < 0.001). The albumin value was statistically significantly lower on the fourth day than on the first day of the study (2.7 (2.4–3.1) g/L vs. 2.5 (2.2–2.9) g/L, *p* = 0.001).

A significant negative correlation was observed between the delta asprosin level and the delta RF value of the patients (Rho = −0.369, *p*= 0.013) (Figure 3).

There was a moderate correlation between the serum asprosin level of the patients and the received % of the daily energy target (Rho = 0.345, *p* = 0.027) (Figure 4).

The correlations between the serum asprosin value and the severity of illness and biochemical parameters of patients on both study time points are summarized in Table 4.

A negative correlation was determined between albumin and prealbumin levels and the first day and delta asprosin levels (*p* < 0.05).

## 4. Discussion

To the best of our knowledge, this prospective study is the first to investigate the serum asprosin value and its relationship between muscle mass and nutritional adequacy in critically ill older adult patients. Most of the participants had increased serum asprosin levels upon study admission. On the fourth day after enteral nutrition support initiation, the serum asprosin concentration of the study sample significantly decreased compared to the first day of the study. There was a significantly negative correlation between the delta asprosin value and the delta RF of patients. Besides, the delta asprosin value was significantly correlated with the received percentage of energy intake from daily energy requirements.

Almost all patients had elevated asprosin levels on the first day of the study. A significant decrease in asprosin levels was observed in our patients after four days of enteral nutrition. We think that the reason for the high first-day asprosin level in patients is malnutrition in adult patients and/or high insulin resistance developing in critical illness. The mean age of our patients was high, and 70.0% were at risk of malnutrition in our study. The main concern in the elderly, especially the very elderly and those with multiple comorbidities, is reduced food intake and weight loss. Malnutrition in elderly patients delays recovery in both acute and chronic diseases and increases morbidity and mortality [22,23]. In response to starvation with a low-intake diet, asprosin is released from white adipose tissue and transported to the liver to mediate glucose release into the bloodstream. Additionally, asprosin is abundantly expressed in human skeletal muscle-derived mesoangioblasts, suggesting that the musculoskeletal system may play a role in regulating asprosin expression [24]. In a cross-sectional study by Hu et al., 46 patients with anorexia nervosa were included. It was found that these patients had a statistically significant increased plasma asprosin level compared to healthy controls [25]. Providing adequate nutritional support in critically elderly patients may be a key method in optimizing increased asprosin levels.

It was shown that insulin resistance in intensive care patients is considerably higher than in healthy patients [10]. In a cross-sectional study conducted by Goodarzi et al., it was reported that the serum asprosin level was statistically significantly positively correlated with Hba1c, HOMA-IR, and insulin levels in patients with a type 2 diabetes mellitus diagnosis and nephropathy [26]. Similarly, the first-day glucose values of our patients were higher than the glucose values after four days of feeding.

Factors including systemic inflammation, decreased peripheral blood flow, inactivity, insulin resistance, and decreased food intake might cause significant reductions in muscle mass in severely ill patients hospitalized in intensive care units. Malnutrition, depending on the negative nutritional balance between what is necessary for the patients and what they receive, is reliant on decreased muscle mass and functionality, which is considered common among ICU patients. Thus, the correct nutritional diagnosis of these patients is critical to support adequate dietary maintenance. Nevertheless, nutritional evaluation is challenging in intensive care units, particularly when monitoring nutritional status. Ultrasonography is a portable, non-invasive bedside method that may specify and measure skeletal muscle and has been used as a supportive examination tool to provide nutritional diagnostics. The ability to detect short-term changes by allowing serial measurements is one of the most advantageous aspects of ultrasonography compared to other anthropometric measurement instruments. The ultrasound examination of rectus femoris muscle thickness has been reported to be used in the monitoring of nutrition [27,28].

In the study of Duerrschmid et al., which experimented with mice, wild-type mice and mice with truncated mutations in the FBN1 gene were provided a high-calorie, high-fat diet. Mice with truncated mutations in the FBN1 gene had less fat and muscle content than wild-type mice. This study demonstrates that asprosin, encoded by FBN1, has an impact on nutrition and muscle mass [29]. One of our hypotheses in our study was that insulin resistance, being very common in intensive care units, can be improved as adequate nutrition is provided, and the increase in the asprosin level may be effective in this condition. Since we did not measure insulin resistance, we cannot clarify this.

This muscle wasting also adversely affects the clinical outcomes of the patients. Quadriceps’ muscle thickness is used to evaluate nutritional interventions in critically ill patients in the intensive care unit [30]. The present study demonstrated a negative correlation between the change in quadriceps’ muscle thickness and asprosin after four days of nutrition. In the current literature, it has been reported that the musculoskeletal system effectively regulates the level of asprosin [24,31]. Du et al. conducted a cross-sectional study of 120 cancer patients. A statistically significant positive correlation was found between the serum asprosin level and body fat mass in these patients [32]. Moreover, increased levels of asprosin may accelerate the reduction in muscle mass in critically ill elderly patients.

In intensive care units, giving sufficient nutritional substances and using them in the anabolic process positively contributes to the course of the disease. We determined a negative correlation between the serum asprosin level of our patients and the percentage of patients reaching the target energy.

The present study concluded a negative correlation between prealbumin and albumin with the asprosin value on the first day. Prealbumin and albumin values are frequently used as biomarkers of adequate nutrition. However, serum albumin and prealbumin levels are affected by many factors [33,34].

Biochemical measures are beneficial to obtain in the ICU setting. Nonetheless, improvements in such parameters are not consistently related to improvements in outcomes when controlled for illness severity. There may be several reasons for these limitations [35]. The significant fluid shifts in critical illness can impact the serum concentrations of the most commonly used biochemical indicators. Visceral or “hepatic” proteins, including albumin and prealbumin, are affected by the acute phase response, independent of nutrition status or nutrition input [36].

For example, the prealbumin level usually falls at the beginning of the ICU admission even when nutrition support has been entirely implemented, and the level may improve as the acute phase response decreases, even if the patient has not received adequate nutrition or has continued to lose weight. However, several studies have suggested that if the acute phase response is reasonably stable, the prealbumin levels then correlate with nutrition intake, but perhaps not the outcome. Prealbumin levels cannot be measured at all hospital laboratories [37,38]. Nonetheless, the serum albumin level is routinely measured in most hospitals and is a robust prognostic indicator even in critical illness, but it has a long half-life and does not correlate to significant alterations in nutrition input, making it less useful as a parameter for sequential monitoring of nutrition progress [39].

Malnutrition is a very common and vital issue in intensive care. There are no good markers to assess rapidly developing muscle wasting in patients with fractures. The prealbumin and albumin we used in our routine are affected by many parameters.

Our study suggests that asprosin, a new adipokine, can be used to monitor adequate nutrition and muscle loss. However, a larger number of patients and further studies are needed.

## 5. Limitations

The limitations of our study are that it is single-centered, and the number of patients is small. Our study findings included only elderly critically ill patients. This limits generalizability in critically ill patients. If we had also evaluated insulin resistance in our patients, we could better explain the pathophysiology.

Another limitation of our study was the inability to measure inflammatory and anti-inflammatory cytokines in patients due to cost. If we could measure the cytokine values, we could see the effect of inflammation on nutrition and asprosin more clearly.

Evaluation with ultrasonography for a longer time would have yielded more precise results to better evaluate the response of muscle mass in response to feeding in patients. The mitochondrial evaluation was not performed for the asprosin level in our study. More reliable results could have been obtained with this evaluation.

## 6. Conclusions

Nearly all the critically ill elderly patients had elevated serum asprosin levels. Serum asprosin levels decreased in those patients who received enteral nutritional support and ICU treatment. In this study, there was a negative correlation between the serum asprosin level and delta lean muscle mass. Additionally, the serum asprosin level correlated with nutritional adequacy. For future investigations, whether the serum asprosin level can be used as a biomarker in evaluating the adequacy of nutritional interventions in critically ill patients should be evaluated with larger sample sizes. The relationship between the asprosin level and ICU-acquired weakness should be clarified.

## Figures and Tables

**Figure 1 healthcare-11-00732-f001:**
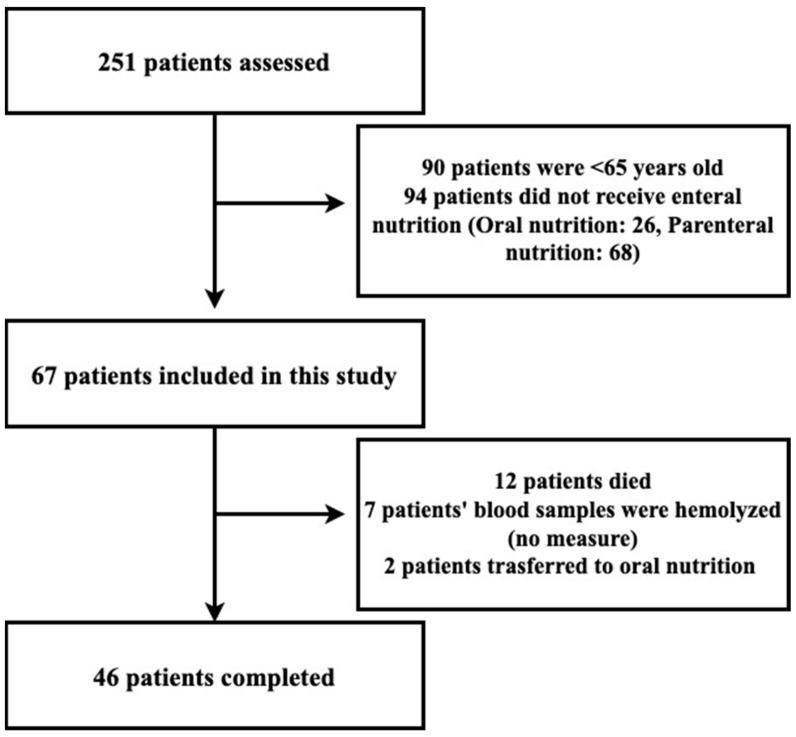
Flowchart of the study.

**Figure 2 healthcare-11-00732-f002:**
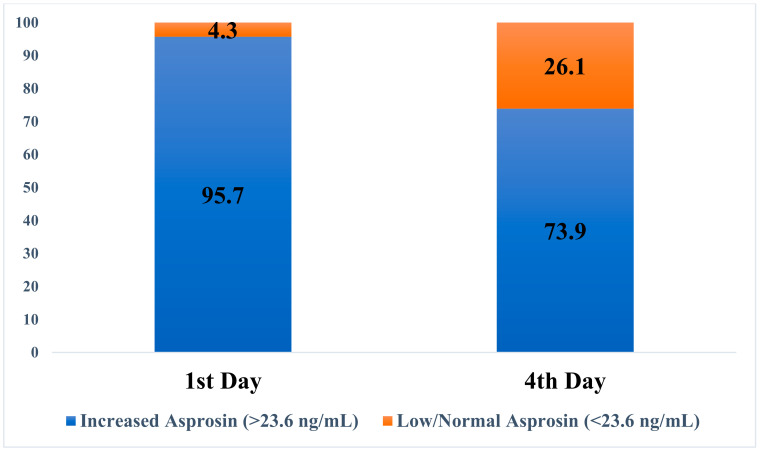
The distribution of patients with high/low–normal asprosin levels on the first and fourth days after EN initiation.

**Figure 3 healthcare-11-00732-f003:**
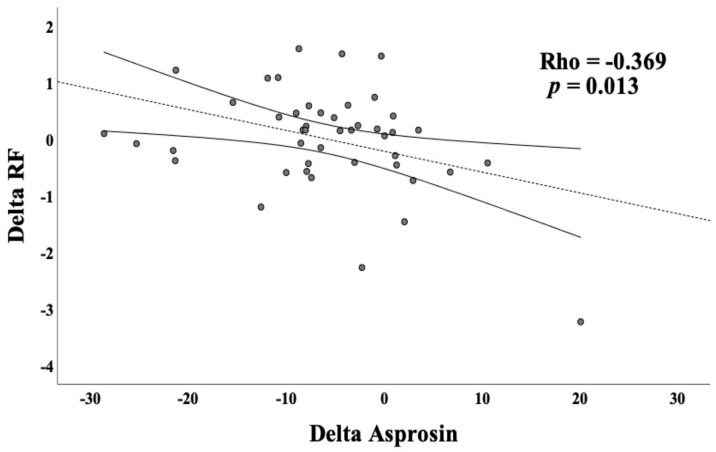
The Spearman correlation between the delta RF and the delta asprosin value.

**Figure 4 healthcare-11-00732-f004:**
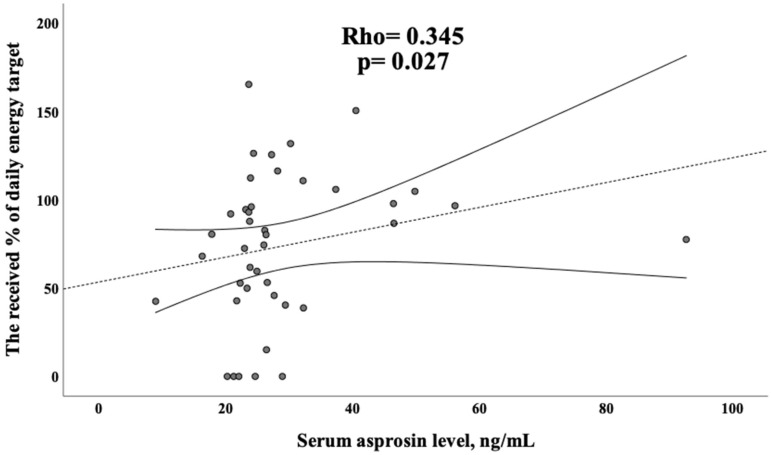
The Spearman correlation between the serum asprosin level and the received % of the daily energy target.

**Table 1 healthcare-11-00732-t001:** Demographic and clinical characteristics of the study participants.

Variables	Total (*n* = 46)
**Age (years) ± SD**	72 ± 6
**Gender, *n* (%)**	
Male	25 (54.3)
Female	21 (41.7)
**BMI, median (IQR)**	22.0 (20.9–29.0)
**Reason for ICU admission, *n* (%)**	
Respiratory failure	8 (17.4)
Sepsis/septic shock	12 (26.1)
Cerebrovascular disease	8 (17.4)
Metabolic reasons	17 (37.0)
Post-op	1 (2.2)
**Comorbidity disease, *n* (%)**	
Diabetes mellitus	16 (34.8)
Hypertension	29 (63.0)
Chronic obstructive pulmonary disease	12 (26.1)
Cardiac failure	10 (21.7)
Chronic renal failure	8 (17.4)
Dementia	10 (21.7)
Other (…)	17 (37.0)
**APACHE II score, ± SD**	19.8 ± 6.98
**SOFA score, ± SD**	
Day 1	6.8 ± 3.43
Day 4	7.3 ± 4.17
**Charlson comorbidity index, median (IQR)**	6.0 (4.0–8.0)
**Baseline Glasgow score, median (IQR)**	10.0 (3.0–13.0)
**Malnutrition at risk, *n* (%)**	32 (70.0)
**Daily energy requirement (kcal/day), ±SD**	1540 ± 193.0
**Daily energy intake (kcal), ±SD**	
Day 1	530 ± 371.3
Day 2	931 ± 574.5
Day 3	1023 ± 560.4
Day 4	1062 ± 622.9
**The received % of daily energy target, ±SD**	
Day 1	34.9 ± 24.8
Day 2	61.4 ± 38.7
Day 3	67.7 ± 37.4
Day 4	74.0 ± 42.0
**Daily protein intake (g/kg/day), ±SD**	
Day 1	0.4 ± 0.27
Day 2	0.7 ± 0.44
Day 3	0.8 ± 0.42
Day 4	0.8 ± 0.47
**The received % of daily protein target, ±SD**	
Day 1	35.8 ± 23.54
Day 2	65.3 ± 40.84
Day 3	73.1 ± 41.22
Day 4	79.3 ± 46.67
**Need for MV support, *n* (%)**	21 (45.7)
**Need for RRT, *n* (%)**	21 (45.7)
**Length of ICU stay (day), median (IQR)**	11.5 (9.8–18.5)
**Length of hospital stay (day), median (IQR)**	19.0 (13.0–31.8)
**Mortality, *n* (%)**	29 (63.0)

APACHE II: Acute Physiology and Chronic Health Evaluation, BMI: body mass index, SOFA: Sequential Organ Failure Assessment, IQR: interquartile range, MV: mechanical ventilation, RRT: renal replacement therapy.

**Table 2 healthcare-11-00732-t002:** Serum asprosin concentrations and RF values of participants on the first and fourth study days.

	Day 1	Day 4	Delta (Δ)	*p*
**Asprosin value, ng/mL**	31.8 (27.4 to 38.1)	26.1 (23.4 to 32.3)	−5.77 (−9.21 to 0.28)	**<0.001**
**RF, cm^2^**	1.68 (1.35 to 2.07)	1.82 (1.38 to 2.13)	0.15 (−0.43 to 0.46)	0.196

**Table 3 healthcare-11-00732-t003:** The laboratory values of the patients on the first and fourth study days.

	Day 1	Day 4	*p*
Glucose (mg/dL)	143 (110–194)	125 (103–170)	**<0.001**
AST (IU/L)	24 (15–35)	22 (13–38)	0.651
ALT (IU/L)	15 (8–24)	16 (9–25)	0.433
LDH (U/L)	294 (237–395)	267 (200–389)	0.040
Albumin (g/dL)	2.7 (2.4–3.1)	2.5 (2.2–2.9)	**0.001**
Prealbumin (g/L)	0.1 ± 0.04	0.1 ± 0.05	0.051
CRP (mg/L)	72.3 (36.7–152.5)	77.6 (30.9–139.5)	0.666
Procalcitonin (µg/L)	0.49 (0.21–1.96)	0.60 (0.14–3.30)	0.984

AST: aspartate aminotransferase, ALT: alanine aminotransferase, LDH: lactate dehydrogenase, CRP: C-reactive protein.

**Table 4 healthcare-11-00732-t004:** The Spearman correlation analysis between serum asprosin levels and severity of illness scores and some biochemical parameters.

		Asprosin Value, ng/mL		
		Day 1	Day 4	Delta
**APACHE II**		−0.160	-	−0.050
**mNUTRIC score**		−0.063	-	0.102
**Charlson comorbidity index**		0.151	-	0.012
**SOFA**	Day 1	0.067	-	0.089
	Day 4	-	0.142	0.133
**Glucose**	Day 1	−0.052		−0.060
	Day 4	-	−0.067	
**AST**	Day 1	−0.088		−0.078
	Day 4		−0.075	
**ALT**	Day 1	−0.082		−0.056
	Day 4		−0.111	
**LDH**	Day 1	−0.007		−0.096
	Day 4		−0.026	
**Albumin**	Day 1	−0.315 *		−0.525 *
	Day 4		−0.002	
**Prealbumin**	Day 1	−0.334 *		−0.322 *
	Day 4		−0.267	
**CRP**	Day 1	−0.106		0.077
	Day 4		0.154	
**PCT**	Day 1	−0.001		0.204
	Day 4		−0.120	

*: *p* < 0.05.

## Data Availability

The data that support the findings of this study are available from the corresponding author upon reasonable request.

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
