# Peer review of "Importance of Asprosin for Changes of M. Rectus Femoris Area during the Acute Phase of Medical Critical Illness: A Prospective Observational Study"

_healthcare, 2023, doi:10.3390/healthcare11050732_

Round 1

Reviewer 1 Report

The authors evaluate a very ill and aged population for plasma asprosin levels, and compare to a variety of other lab studies and measure of muscle mass. The major finding is that asprosin levels are high in this population and there is a correlation to RF muscle area.

The manuscript generally reads well and the approach is clearly written.

Major points:

1) Figure 2 (page 6), demonstrates increased asprosin level. Table 1 shoes comorbidities, including 34.8% of the population having diabetes.

a) Please state the percent of population receiving insulin, in table 1.

b) from my review of methods, page 2 (line 92) to page 3 (line 99) the collection of asprosin is not clearly mentioned to be at a fasting state, but may be at a fed state depending on the clinical situation. If this is correct, please add a statement of caution regarding interpretation of nonfasting asprosin levels, as levels are dependent on fasting vs fed state.

2) Page 7, Figures 3 and 4 show Correlations with delta asprosin value.

a) The Scales on X-axis between the 2 figures are different. There appears to be one individual that has a delta asprosin of ~ +60 units who is not shown on the delta RF figure. Please adjust scales for consistency.

b) RF area is thought to be a good marker of nutrition. Can you please share/show the correlation between RF area and nutrition either in text or supplementary figure, as this will help the reader have confidence in RF area measurement validity.

There are some suggestions to make the manuscript

Author Response

We would like to thank you for your careful reviewing, helpful comments, and constructive suggestions, which has significantly improved the presentation of our manuscript. We have carefully considered all comments from the reviewer and revised our manuscript accordingly. The manuscript has also been double-checked, in the following section, we summarize our responses to each comment from the reviewer. We believe that our responses have well addressed all concerns from the reviewer. We hope our revised manuscript can be accepted for publication.

Major points:

1) Figure 2 (page 6), demonstrates increased asprosin level. Table 1 shoes comorbidities, including 34.8% of the population having diabetes.

  1. a) Please state the percent of population receiving insulin, in table 1.

It is stated in the result section that all diabetes mellitus patients include insulin therapy.

  1. b) from my review of methods, page 2 (line 92) to page 3 (line 99) the collection of asprosin is not clearly mentioned to be at a fasting state, but may be at a fed state depending on the clinical situation. If this is correct, please add a statement of caution regarding interpretation of nonfasting asprosin levels, as levels are dependent on fasting vs fed state.

 In our intensive care unit, feeding is interrupted at 11 am for all patients receiving enteral nutrition. Blood was drawn in the morning fasting before feeding was reopened.(line 193)

2) Page 7, Figures 3 and 4 show Correlations with delta asprosin value.

  1. a) The Scales on X-axis between the 2 figures are different. There appears to be one individual that has a delta asprosin of ~ +60 units who is not shown on the delta RF figure. Please adjust scales for consistency.

I am very sorry that I wrote the word serum instead of delta in the chart, thank you very much for your attention.

Actually, it looks like 60 because it's serum asprosin level.

  1. b) RF area is thought to be a good marker of nutrition. Can you please share/show the correlation between RF area and nutrition either in text or supplementary figure, as this will help the reader have confidence in RF area measurement validity.

It has been tried to explain in lines 353-368.

Reviewer 2 Report

The authors analyzed the importance of asprosin for changes in the M. rectus femoris region during the acute phase of critical medical illness in a prospective observational study and noted a negative correlation between serum asprosin level and delta lean muscle mass.

The topic is beautiful and remarkable, but I also have a few comments regarding the merit content of the article:

1. Overall, the manuscript is written in poor English and requires language improvements.

2. 211-212. In the lines, the authors said that increased serum asprosin levels may accelerate the reduction of muscle mass in critically ill elderly patients. However, they did not provide references. They could not explain this.

 3. The relationship between the serum asprosin level of the patients in lines 214-220 and the percentage of reaching the target energy, and the relationship between prealbumin/albumin and asprosin could not be fully explained.

Author Response

We would like to thank you for your careful reviewing, helpful comments, and constructive suggestions, which has significantly improved the presentation of our manuscript. We have carefully considered all comments from the reviewer and revised our manuscript accordingly. The manuscript has also been double-checked, in the following section, we summarize our responses to each comment from the reviewer. We believe that our responses have well addressed all concerns from the reviewer. We hope our revised manuscript can be accepted for publication.

  1. Overall, the manuscript is written in poor English and requires language improvements.

The manuscript was edited in English.

  1. 211-212. In the lines, the authors said that increased serum asprosin levels may accelerate the reduction of muscle mass in critically ill elderly patients. However, they did not provide references. They could not explain this.

In the discussion section, the desired explanation was tried to be made in 370-377 In the lines.

  1. The relationship between the serum asprosin level of the patients in lines 214-220 and the percentage of reaching the target energy, and the relationship between prealbumin/albumin and asprosin could not be fully explained.

In the discussion section, the desired explanation was tried to be made in 411-433 In the lines